# Active colloids as mobile microelectrodes for unified label-free selective cargo transport

Alicia M. Boymelgreen[1], Tov Balli[1], Touvia Miloh[2] & Gilad Yossifon[1]

Utilization of active colloids to transport both biological and inorganic cargo has been widely examined in the context of applications ranging from targeted drug delivery to sample analysis. In general, carriers are customized to load one specific target via a mechanism distinct from that driving the transport. Here we unify these tasks and extend loading capabilities to include on-demand selection of multiple nano/micro-sized targets without the need for pre-labelling or surface functionalization. An externally applied electric field is singularly used to drive the active cargo carrier and transform it into a mobile floating electrode that can attract (trap) or repel specific targets from its surface by dielectrophoresis, enabling dynamic control of target selection, loading and rate of transport via the electric field parameters. In the future, dynamic selectivity could be combined with directed motion to develop building blocks for bottom-up fabrication in applications such as additive manufacturing and soft robotics.

[1] Faculty of Mechanical Engineering, Micro- and Nanofluidics Laboratory, Technion – Israel Institute of Technology, Haifa 32000, Israel. [2] School of Mechanical Engineering, University of Tel-Aviv, Tel-Aviv 69978, Israel. Correspondence and requests for materials should be addressed to G.Y. (email: yossifon@technion.ac.il)

I n exploring the application of synthetic active colloids towards cargo transport and delivery, we note that the carrier delivery system (propulsion and directed control) and cargo manipulation (loading and release) are most commonly approached as two disparate problems. In terms of delivery, the freedom of active colloids to travel along individual pathlines constitutes a significant advantage vis-a-vis particles driven by externally controlled fields or gradients which move phoretically, i.e., migrating en-masse in an externally dictated direction[1]. A second desirable characteristic of most active systems is that the colloids move autonomously[2,3], driven by self-propulsive mechanisms such as mechanical swimming or catalytic reactions[2,4]. However, such propulsion is often accompanied by challenges including directed motion, the non biocompatibility of common fuels such as hydrogen peroxide and the finite life of the micromotors resulting from fuel consumption[2,3]. Although creative solutions to these challenges which maintain autonomy are being explored, such as fuel-free water-driven catalytic reactions[5] and boundary guidance[6], in many cases, a cost-benefit analysis suggests it is more practical to resort to integration of secondary mechanisms to perform these tasks, such as the use of external fields (e.g., electric[7] or magnetic[3,8]) for guidance or even switch entirely to fuel-free, externally powered (magnetic, electric, light and ultrasonic) propulsion mechanisms[9], which may not be active.

Uniquely, under the application of a uniform AC electric field, metallodielectric Janus particles (where one hemisphere is conducting and the other dielectric) respond as active colloids[10,11]—travelling along individual pathlines and exhibiting group behaviour, despite the external nature of the applied field. This distinctive characteristic arises from the propulsive mechanism—either induced-charge electroosmosis[12,13] or self-dielectrophoresis[11] (sDEP)—being produced on the individual particle level rather than via an externally applied global gradient. When compared with challenges facing other active colloids, we note that the process is fuel free; in fact, mobility is greatest in aqueous electrolytes, including water[12]. Moreover, the same AC field used to dictate the frequency-dependent mobility can be used to steer the carrier and obtain directed motion via a feedback control loop[14] or by producing vortices, which guide the particles' path via hydrodynamic confinement[7]. In applications where the frequency range is restricted, one could integrate a secondary steering mechanism such as magnetic guidance[15] or physical boundaries[6].

Turning our focus to cargo manipulation, we first note that within the current literature, the mechanisms for cargo selection can be grossly classified into two general categories: application of external fields and surface functionalization. The former has the advantages of on-demand pickup and release, and may also be used for steering, but to date has been limited to magnetically labelled cargo[8,16–20]. Alternatively, a broad range of cargo can be transported by surface functionalization, e.g., hydrophobic surface interaction[21], biomolecular reactions[22,23] and electrostatic interaction[23], but these mechanisms are restricted to a specific and predefined target. Furthermore, efficiency is limited by the rate of reaction and release of the cargo (if possible) is complicated[24].

In contrast, dielectrophoresis (DEP), the frequency-dependent mechanism used in the current work to load and release cargo, offers a label-free method to selectively manipulate a broad range of both organic and inorganic matter[25], where the frequency dependence also enables the unique flexibility to change the choice of target in situ. The working principle relies on the fact that in the presence of the non-uniform field, depending on whether a target is more or less polarizable than the surrounding medium, it will either be attracted to (positive dielectrophoresis, pDEP) or repelled from (negative dielectrophoresis, nDEP) the

region of high field strength. The pDEP–nDEP transition is dictated by the Clausius–Mossotti factor, which is a function of the relative material properties (complex permittivities) of the target and suspending fluid, and, in some cases, such as small dielectric particles, the colloidal geometry as well[26,27]. As the material properties are complex, the Clausius–Mossotti factor is frequency dependent. Previously, dielectrophoresis has been widely used to separate, concentrate and characterize a broad range of both organic and inorganic matter in microfluidic chambers where the requisite non-uniformity of the field gradients is attained by pre-patterning of the microfluidic chamber[25,28]—either with active or floating electrodes or insulating geometries.

Here we show that the field gradients necessary to manipulate matter via dielectrophoresis can also be induced at the surface of a polarizable, freely suspended colloid, enabling the development of a unified micro cargo carrier that can both attract a broad range of cargo and where the choice of target, its accumulation, transport and release are singularly controlled via the external electric field. Although these gradients may also be observed around homogeneous metallic spheres[29], within this work, we have chosen to focus on the Janus sphere, as it has the advantage of translating as an active colloid and the non-uniformity of the sphere's composition results in stronger three-dimensional (3D) electric field gradients, leading to a more pronounced effect. The findings hinge on the recognition that as well as driving the active colloidal mobility of the Janus sphere, the polarization of the metallic hemisphere under the applied electric field essentially transforms the particle into a mobile floating electrode, where the finite size of the colloid results in a localized disturbance to the electric field distribution, represented by the rainbow gradients in Fig. 1a. It is demonstrated that the resultant localized non-uniform field around the mobile microelectrode can be exploited to selectively trap and release targets (Fig. 1b,c) via a dielectrophoretic force[30], wherein by altering the frequency of the electric field, the particle may either be attracted to (pDEP) or repelled from (nDEP) regions of high field strength according to the target's individual Clausius–Mossotti factor. Here we characterize the localized induced gradients at the surface of the Janus sphere and demonstrate that loading and transport are functions of cargo and carrier geometries and the frequency and intensity of the applied field. It is emphasized that the dielectrophoretic manipulation described here, around a mobile, freely suspended colloid whose position can be dynamically altered, is distinct from that studied extensively in the literature where the pre-patterning of the microfluidic chamber results in a trapping site, which is a fixed, inherent property of the system that must be optimized at the time of fabrication.

## Results

**Target loading via the electric field gradient induced around a Janus sphere.** In order to define the dependence of the cargo loading on the applied electric field, we note that a target will be trapped when the attracting dielectrophoretic potential, generated by the non-uniformity of the induced field around the Janus sphere, exceeds the thermal potential (Brownian motion) of the target such that (see Supplementary Note 1 and Eq 1)

$$V_{RMS}^2 \sim \frac{C}{|\frac{\partial \tilde{\chi}}{\partial x_2}|}; C = \frac{\Delta k_B T L^2}{\pi \varepsilon_f b^3 \text{Re}\{K_{CM}\}} \tag{1}$$

where $V_{RMS}$ is the root mean square of the applied field, $\frac{\partial \tilde{\chi}}{\partial x_2}$ is the dominant normalized derivative of the induced potential around the colloid, $k_B$ is the Boltzmann constant, $T$ is temperature, $L$ is the distance between electrodes, $\varepsilon_f$ is the permittivity of the fluid, $b$ is the target radius, $\text{Re}\{K_{CM}\}$ is the real part of the

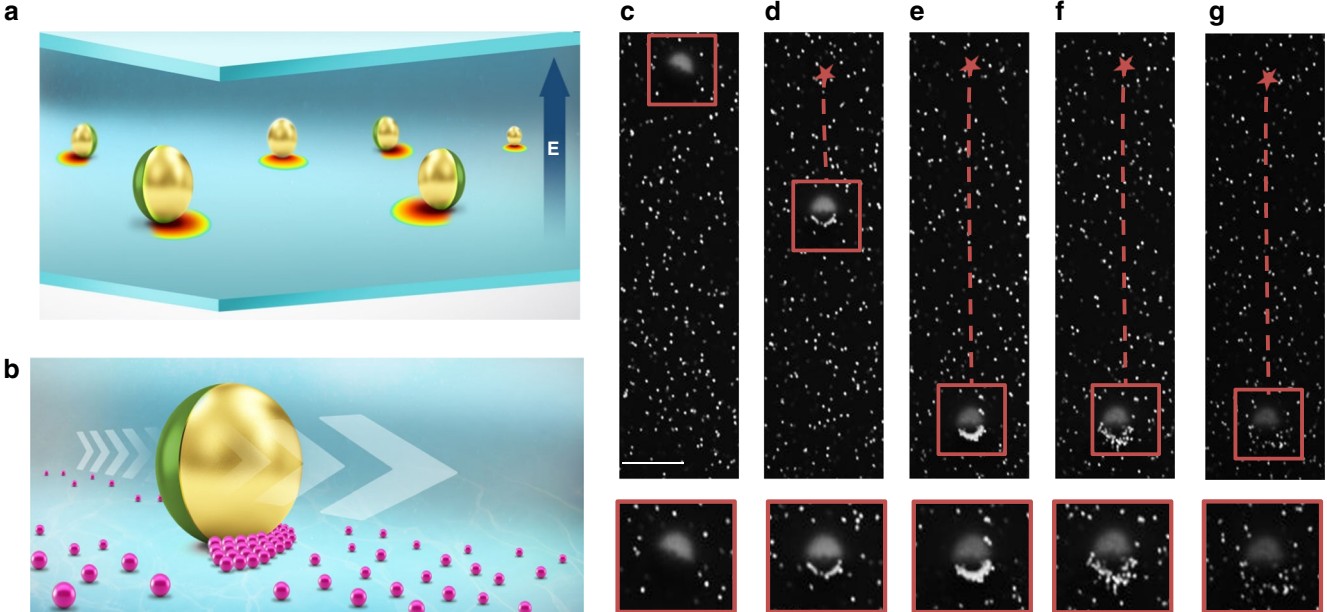

**Fig. 1** Mobile microelectrode applied to cargo transport. **a** Schematic of the general concept of mobile microelectrode; localized gradients are induced around polarizable surface such that the spatial distribution of the field gradient in the system may be varied by manipulating the position of the Janus sphere; **b** Schematic illustrating concept of cargo transport; Janus particle picks up cargo, as it translates and transports it to a secondary location. **c**–**g** Microscope images showing accumulation, transport and release of 300 nm target by a 15 μm Janus particle carrier controlled by the frequency of the applied electric field.Video available in real time as Supplementary Video 1. **c**–**e** At 100 kHz, the 300 nm target accumulates at the metallic hemisphere under attractive pDEP. **f**, **g** At 2 MHz the target experiences a negative nDEP force and is ejected from the carrier surface. Scale bar illustrated in **c** is 10 μm

Clausius–Mossotti factor and $\Delta$ represents the ratio of the dielectrophoretic to thermal potentials[31], left here as a single fitting parameter. For the current experimental setup, $L = 120$ μm, $k_B = 1.38 \times 10^{-23}$ m$^2$ kg s$^{-2}$ K$^{-1}$, $T = 298$ K, $\varepsilon_f = 80 \cdot 8.85 \times 10^{-12}$ Fm$^{-1}$, $b = 150$ nm, Re$\{K_{CM}\} \sim 1$ (high frequency, pDEP) such that $C = 8\Delta$ V$^2_{RMS}$(see Supplementary Note 1 and Eqs 1–8 for details).

The precise distribution of the electric field, induced around the mobile floating electrode is dependent on the geometry and material properties of the particle. Envisioning the polarized colloid as a source, one can theoretically derive the induced distribution for any geometry in terms of a multipole expansion, which decays to zero in the far-field (satisfying the Laplace equation) and where the magnitudes of the included terms are prescribed by the electrostatic boundary conditions at the colloid surface[32]. In the practical case, where the colloid is often in close proximity to the wall[12,29], it is necessary to also consider the images of the multipole expansion, as the interaction of the induced field with the channel boundaries can dominate—as in the current setup where gradients below the Janus sphere are significantly larger than those on top (Fig. 2a). For a metallodielectric Janus sphere, a good approximation of $\tilde{\chi}$ can be obtained by considering the sum of a point dipole and multipole[33], where the latter accounts for the symmetry breaking (see Supplementary Note 1, Eq. 5).

For the present case of a metallodielectric Janus sphere, qualitative examination of the 3D trapping around a stagnant Janus particle adjacent to a wall (Fig. 2b–d) illustrates that in the current setup, the strongest gradients are formed in the nanometer-sized gap between the particle and wall (Fig. 2d), with smaller gradients also observed at the upper interface between metallic and dielectric hemispheres (Fig. 2c).

Focusing on the region of maximum field strength, in parts e–g, we observe that as the sphere size increases, the minimal critical voltage below which no targets are trapped also decreases. Correspondingly, as the voltage is increased, larger spheres are able to attract more cargo— reflected in increased trapping area —as for a given voltage, the critical field corresponding to the dominance of DEP over thermal forces occurs farther from the center of the Janus particle (JP). Noting that the area, $A$, of trapped target (fluorescent region in the inset of Fig. 2e) is approximately hemispherical, we can express $A$ in terms of the non-dimensional parameter $\eta = \left|\frac{x_1}{a}\right|$ (where $a$ is the radius of the Janus sphere) such that $\eta^2_{max} = \frac{x^2_{1,max}}{a^2} \sim \frac{2A + \pi\zeta^2}{\pi a^2}$. Here, $\zeta$ corresponds to the geometrically dictated minimum distance from the center of the particle that can be reached by the target (denoted by a red line in the schematic inset of Fig. 2f). In the limit of $h/a \sim 1$ (i.e., the gap between the centre of the Janus sphere and wall $(h)$ is very small with respect to the radius $(a)$), $\zeta = 2\sqrt{ba}$, so that $\eta|_\zeta = \eta_{min} = 2\sqrt{b/a}$.

In Fig. 2g, we demonstrate that the relationship between $\eta$ and $V$ is well-described by the theoretical curve (solid black line) obtained for $\frac{\partial\tilde{\chi}}{\partial x_2} \sim 2\left(\frac{P_1(2-\eta^2)}{(1+\eta^2)^{5/2}} + \frac{6P_2\eta(4-\eta^2)}{(1+\eta^2)^{7/2}}\right)$ (Supplementary Note 1, Eq. 9), where the saturation of $\eta_{max}$ at slightly lower voltages than theoretically predicted is likely due to $h/a > 1$ and may also be related to particle–particle interactions. For low voltages, a simpler scaling (solid black line) can be obtained using the linear approximation $C/|\partial\tilde{\chi}/\partial x_2| \sim 0.3(1 + 7\eta)$ (Supplementary Note 1, Eq. 11); yielding good agreement for the critical applied voltage at which the first target is trapped at $\eta_{min}$ (Fig. 2f). For both the full expression and approximate solutions, similar fitting parameters of $\Delta = 8,8.6$ respectively are obtained, corresponding well with the general rule of thumb that requires the DEP potential to exceed the thermal by an order or magnitude for onset of trapping[31].

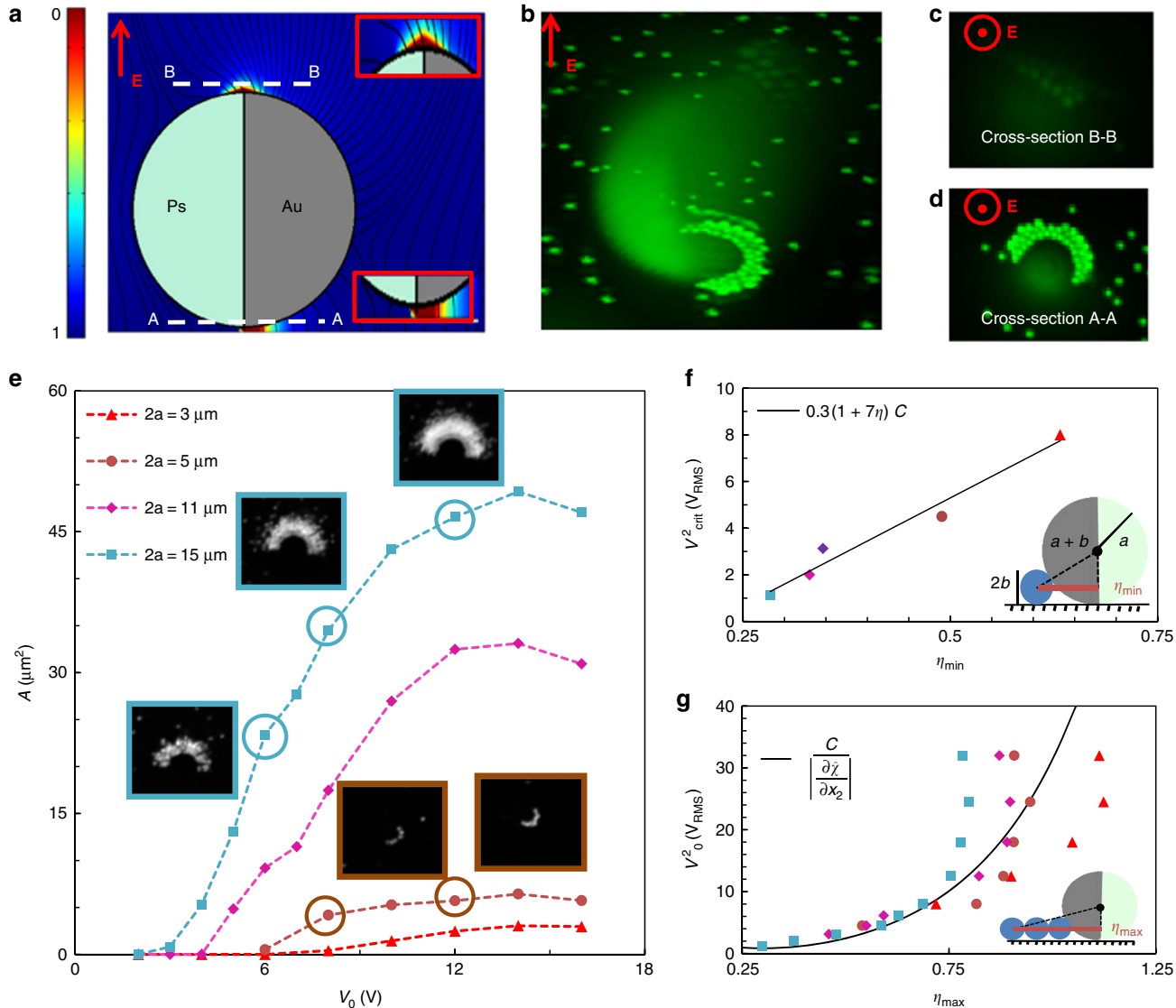

**Fig. 2** Relationship between the strength of the induced field gradient and cargo loading. **a** 2D COMSOL simulation of expected gradients around the Janus sphere. **b**–**d** Experimental observation of the formation of field gradients around a Janus particle: **b** 3D isometric image of trapped targets—indicating regions of high electric field strength—generated from superposition of multiple images taken along the height of the particle; **c** view of trapping at focal plane (B–B) directly above the Janus sphere; **d** view of trapping directly below the particle adjacent to the substrate (A–A). **e** Variation of area of trapped fluorescent 300 nm target around Janus spheres 3, 5, 11 and 15 μm in diameter with respect to the applied voltage. Sample microscopic images of trapping around 5 and 15 μm Janus spheres are shown in the inset. **f** Scaling of the minimum critical voltage required for onset of trapping with the electric field for varying Janus particles sizes $\eta_{min} = |\zeta/a| = 2\sqrt{b/a}$. **g** Scaling of the increased trapping area, $\eta^2_{max} = \frac{x^2_{1,max}}{a^2} \sim \frac{2A + \pi\zeta^2}{\pi a^2}$, as a function of the applied electric field

**Influence of cargo size on loading capacity**. As well as the size of the carrier and the applied voltage, the load capacity of the cargo is characterized by a delicate interplay between the size and shape of the carrier relative to the cargo (Fig. 3). In general, devices relying on DEP as a trapping mechanism are most suited to micron-sized objects—as the force scales with the radius cubed—and the extension to the nanoscale requires complex patterning to obtain the necessary high fields without undesirable high voltages[31,34]. In contrast, the current colloidal setup is optimal for nanotargets, as the smaller particles can penetrate closer to the center of the JP, towards the region of maximum high field strength—as is evidenced by the smaller minimum radii of the area trapped for smaller targets (Fig. 3b). In addition, as larger targets begin to accumulate, they can 'shield' the field intensity,

such that particles further back are not exposed to a field of sufficient strength to induce trapping. This 3D shielding may explain the plateau of the area at high voltages, where despite the increased field, the maximum radius at which particles are trapped does not increase—although for smaller particles we do see a continued increase in intensity, indicating trapping in the $x_2$ direction (i.e., suggesting the formation of more compact multiple layers). This supposition is further supported by the observation that the onset of the plateau occurs at lower voltages as the tracer size increases. The interplay between these geometric constraints and the dielectrophoretic force results in a non-monotonic increase of the trapping area with respect to the target radius, and suggests that for a given target there is an optimal size cargo carrier that can maximize the load carried (Fig. 3a).

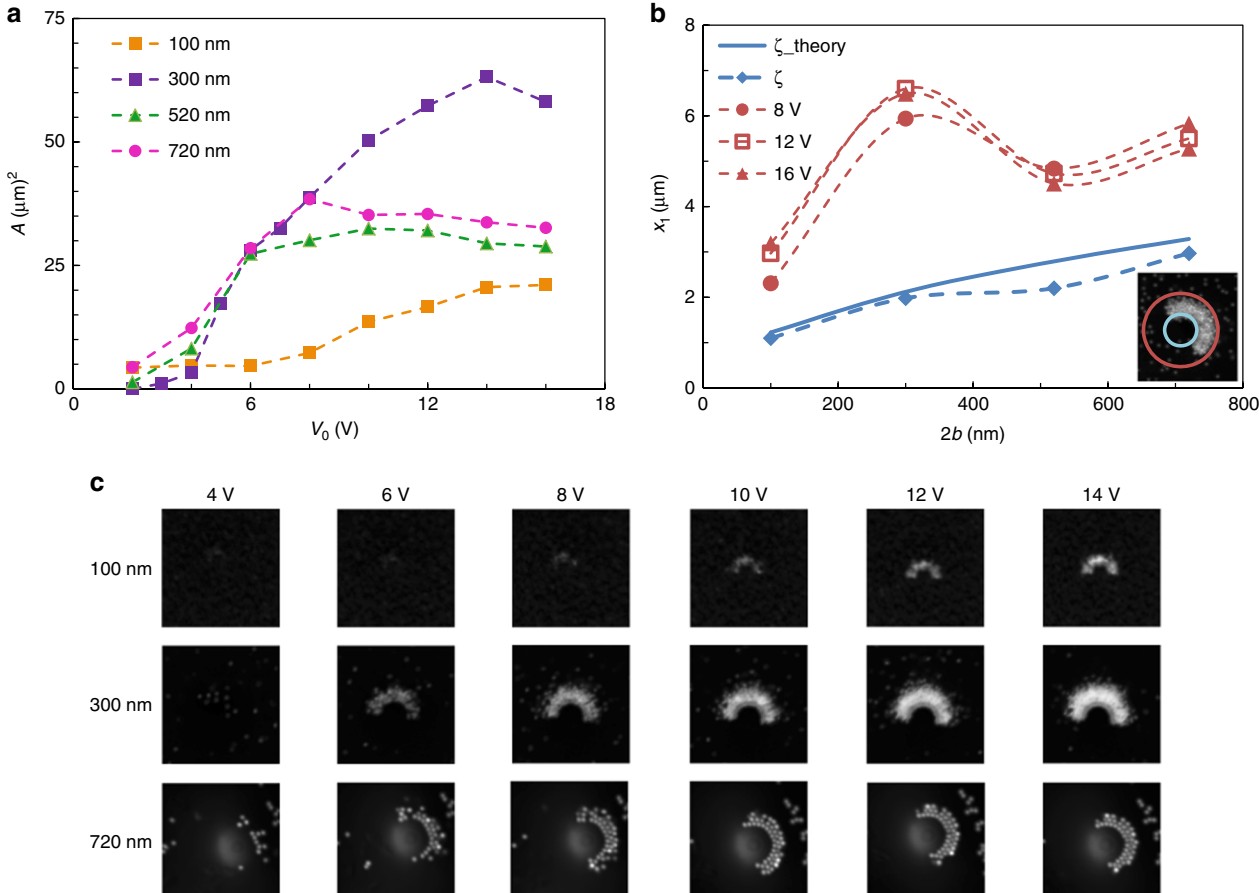

**Fig. 3** Dependency of cargo loading on target size and applied voltage. **a** Plot of area of accumulated targets of varying diameter around a 15 μm Janus sphere as a function of applied voltage. **b** Measured (blue markers) and predicted (solid blue line) minimum radius at which targets accumulate and maximum radius (red markers) for various applied voltages. **c** Microscope images of 100, 300, and 720 nm targets accumulated around a 15 μm Janus sphere for applied fields ranging between 4 and 14 V reflecting a sample of the experimental data used to determine part **a**

**Characterization of frequency enabled selectivity of the cargo carrier.** Although the load is controlled by the applied voltage and carrier geometry, the selectivity of the mobile microelectrode carrier arises from the fact that all materials exhibit a unique frequency response, wherein according to the value of the real part of the individual, frequency-dependent Clausius–Mossotti factor ($K_{CM}$), a target will either be attracted (and trapped) at the region of high field strength (pDEP, Re$\{K_{CM}\}>0$) or repelled from it (nDEP, Re$\{K_{CM}\}<0$). For most applications, satisfactory prediction of the pDEP to nDEP transition, i.e., crossover frequency (COF) at which Re$\{K_{CM}\} = 0$ may be obtained using the simple dipole approximation[30] in which $K_{CM}$ is defined as a function of the relative material dielectric properties (permittivity and conductivity) of the particle and suspending medium[30,35] (Eq. 3 in Supplementary Note 1). In the case of small dielectric particles, such as the targets used in this work, surface conduction can dominate the negligible material bulk conductivity, necessitating the addition of a correction factor, which is a function of surface charge and geometry (size)[26,27] (see Supplementary Note 1, Eq. 3). Provided that the COFs are sufficiently distinct, by aligning the frequency of the applied field to coincide with the pDEP response of a target and nDEP response of a secondary contaminant within the solution, we may dynamically select a desired target from a mixture (Fig. 4a–e).

To demonstrate this concept, we have used a standard quadrupole to experimentally measure the COF of 300 nm and

1 μm polystyrene spheres, suspended in the same solution used in the experiments, at 1 MHz and 200 kHz, respectively. Recognizing that the COF corresponds to Re$\{K_{CM}\} = 0$, we can extract the size-dependent effective conductivity (arising from surface conductance) of each target[26] as a fitting parameter. Using this corrected conductivity value in the expression for $K_{CM}$ (Supplementary Note 1, Eq. 3), we can plot the real part of the Clausius–Mossotti factor as a function of frequency (purple and magenta curves in Fig. 4) and identify the selective frequency range (highlighted in green), wherein the target (here, 300 nm Ps spheres) undergoes pDEP and accumulates at the surface of the JP, while the 1 μm 'contaminant' is repelled by nDEP (Fig. 4d).

Finally, we note that for active Janus colloids, the frequency will also dictate the mode of carrying. At low frequencies, Janus particles translate under induced-charge electrophoresis where strong hydrodynamic flow around the metallic hemisphere propels the Janus particle with its dielectric hemisphere forward[13]. The combination of the induced-charge electro-osmotic flow, which injects from the bottom of the Janus sphere and ejects at the midplane[29], and the location of the tracers at the aft of the particle results in the concentration of the cargo around the midline of the JP (see Fig. 4b). At high frequencies, there is no hydrodynamic flow and the Janus particle moves with its metallic hemisphere forward, likely to be due to the unbalanced gradients of the applied field[11]. In the high-frequency case, where there is no hydrodynamic flow and

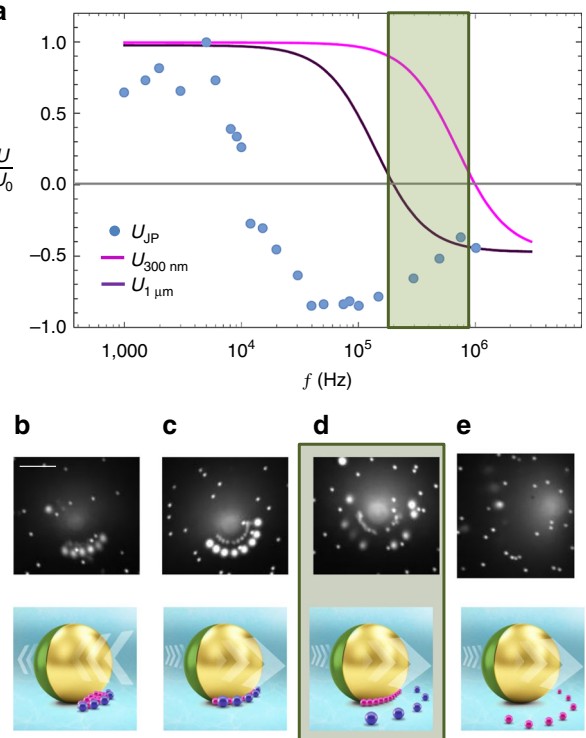

**Fig. 4** Selective cargo assembly and transport. **a** Precise control of the frequency of the applied field enables selective transport (green highlighted region). The desired operating frequency is chosen to align with positive DEP of target (magenta) and negative DEP of the contaminant (purple). The frequency of the applied field also determines the mode of transport—whether by ICEP with Ps hemisphere forwards or sDEP with metallic hemisphere forwards. Operating conditions can be determined by plotting the frequency-dependent mobility (particle velocity normalized by the maximum attainable value $U_0 = 2.4 \, \mu m \, s^{-1}$) JP carrier (blue data points) and the real part of the Clausius–Mossotti factor of target and contaminant (magenta and purple curves). Four frequency domains are observed—each of which correspond with microscope images and schematics in **b–e**. **b** At low frequencies, target and contaminant undergo pDEP (non-selective trapping), whereas JP translates forward under ICEP. **c** Janus particle reverses direction but both target and contaminant still undergo pDEP. **d** Selective trapping—target undergoes pDEP, while contaminant undergoes nDEP. **e** Frequency aligned with nDEP of target for release. Scale bar in **b** is 5 μm

cargo is located fore of the particle, the distribution is hemispherical and remains a monolayer at moderate voltages (Fig. 4c–e and Supplementary Video 2).

## Discussion

In recognizing that electrokinetically driven Janus spheres can function as mobile microelectrodes, capable of manipulating cargo via dielectrophoresis, we have not only successfully unified the transport and loading mechanism of a cargo carrier but also eliminated the need for pre-labelling or surface functionalization of the cargo which in the present configuration can be selectively loaded/unloaded and dynamically varied simply by changing the frequency of the applied field. By integrating methods for directed control into the current system, it is expected that the described cargo carrier could be used in applications such as sample analysis[36], where the carrier can selectively pick up a target and transport it to a secondary chamber for further analysis.

Within the context of dielectrophoretic manipulation, that this floating electrode is mobile essentially represents a paradigm shift relative to the current generation of electrokinetically driven devices built by photolithographic patterning, where the spatial distribution of the field gradient is always a fixed, inherent property of each individual system[28], as the geometry of chamber and electrodes must be predetermined and optimized before fabrication begins. Comparing dielectrophoresis in these two systems, we note that the unfixed electrode is able to trap nanotargets at moderate voltages; without requiring complex (costly) patterning to yield gradients sufficiently strong to compensate for the decrease of the dielectrophoretic force with the radius cubed. Moreover, the active mobility that enables the carrier to travel to the target and transport it to a secondary location simplifies the overall system by eliminating the necessity of a secondary convection mechanism (e.g., pressure-driven flow) commonly employed in high-throughput dielectrophoretic devices to bring the target to the region of high field strength[27,33].

In a global sense, looking beyond the specific applications of dielectrophoretic manipulation and cargo transport, the shift from a pre-patterned chamber with a fixed field gradient to a mobile electrode, where the spatio-temporal distribution of the electric field can be changed in real time by manipulating the position of suspended colloids, essentially represents a step towards the development of top-down to bottom-up fabrication or additive manufacturing techniques in microfluidic systems. Already, similar to the fields of electronics and materials science[37,38] where this transition from top-down to bottom-up has gained much traction, the use of additive manufacturing to build the next generation of microfluidic devices is a burgeoning area of research driven by the expectation of advantages such as decreased cost and increased feature resolution in three dimensions when compared with standard photolithography[39]. By integrating methods for precision-directed control of the current mobile microelectrodes via methods such as electric[14] or magnetic[8,15] steering, one could potentially obtain the aforementioned advantages as well as open up the flexibility to operate in multiple modes, by reconfigurable, dynamic assembly/disassembly. For example, multiple freely suspended colloids could be assembled inside generic microfluidic devices to form necessary microfluidic components such as electrodes on demand. Along with the reduced manufacturing cost and the ability to create 3D and curved geometries, this method puts the control in the hands of the end user who can vary the position in response to experimental conditions in real time. Alternatively, the demonstrated frequency-dependent selectivity and configurability could be used to develop complex building blocks for self-assembly in the design of smart materials[40] or electrically responsive soft actuators towards applications such as soft robotics[41].

## Methods

**Janus particle and solution preparation**. Janus particles 3, 5, 10 and 15 μm in diameter were manufactured by coating green polystyrene spheres (Fluoro-max) with 10 nm Cr followed by 30 nm Au according to the protocol outlined in Ref. [29]. Ps particles (Fluoro-max) of sizes 100, 300, 520, 720 and 1,000 nm were rinsed three times with deionized water, to which a small amount of non-ionic surfactant (Tween 20 (Sigma Aldrich)) was added in order to minimize adhesion to the Indium Tin Oxide (ITO) substrate before being injected into the microfluidic chamber via a small hole at the upper substrate, drilled expressly for this purpose. Based on the size of the polystyrene particles, their concentration ranged between 0.1 and 0.01% (w/v).

**Experimental setup**. The experimental chamber consisted of a 120 μm high silicone reservoir (Grace-Bio) sandwiched between an ITO-coated glass slide (Delta Technologies) and an ITO coated coverslip (SPI Systems) as illustrated in Ref. [29]. Two holes were drilled at the top of the chamber, to ensure the chamber remained

wetted by enabling the addition of electrolyte and colloids into the channel via manual pumping. These holes were surrounded by a reservoir, 2 mm in height filled with solution. The AC electric forcing was applied using a signal generator (Agilent 33250 A) and monitored by an oscilloscope (Tektronix-TPS-2024).

**Microscopy and image analysis**. Observation of the distribution of Polystyrene colloids around the Janus sphere (Figs. 2–4) were performed on a Nikon Eclipse Ti-E inverted microscope equipped with Yokagawa CSU-X1 spinning disk confocal scanner and Andor iXon-897 EMCCD camera. The chamber was placed with the coverslip side down and images were magnified with ×60 oil-immersion lens. Green and red particles were illuminated with lasers of wavelength 488 nm and 561 nm, respectively. To obtain the frequency dispersion in Fig. 1c, the microfluidic chamber was observed under a Nikon TI inverted epi-fluorescent microscope, fitted with a ×10 lens and recorded on a Andor Neo sCOMS camera for a minimum of 30 s at a rate of 5 frames per second. Particle velocities were extracted by tracking particle displacement using PredictiveTracker function in Matlab (https://web.stanford.edu/~nto/software.shtml)) and averaging the velocity over the number of mobile particles. Accuracy of the code was initially verified by comparing the velocities obtained using this method with manual measurement in Image J.

**Numerical simulations**. The numerical simulation used to qualitatively verify the presence of asymmetric electric field gradients arising from the proximity of a Janus sphere near a conducting wall was performed in COMSOL 4.2. A simple two-dimensional geometry was used to model the experimental setup; consisting of a rectangular channel, 80 μm height and 200 μm wide, with a 15 μm diameter circle placed 300 nm above the substrate (this height is based on the observation that the 300 nm tracers do not pass below the JP). The electrostatic equations are solved in the rectangular domain, with the following boundary conditions. At the lower substrate ($y = 0$), a voltage of 5 V is applied while the upper wall is grounded. The edges of the channel are given an insulating boundary condition. The Janus sphere is modelled by applying floating electrode and insulating boundary conditions at the metallic (right) and dielectric (left) hemispheres, respectively.

**Data availability**. The data that support the findings of this study are available from the corresponding author upon reasonable request.

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

## Acknowledgements

We acknowledge US-Israel Binational Science Foundation Grant 2009371, Israel Science Foundation 1945/14 and the RBNI and Gutwirth graduate fellowships. The JP preparation was possible through the financial and technical support of the Technion RBNI (Russell Berrie Nanotechnology Institute) and MNFU (Micro Nano Fabrication Unit). We also thank Dr Sinwook Park for technical assistance.

## Author contributions

A.B. performed/supervised experiments, image and data analysis, numerics, developed scaling analysis and compiled the manuscript. T.B. performed experiments and image analysis. T.M. was responsible for theoretical derivation. G.Y. supervised planning and execution of experiments and assisted in scaling analysis and numerics. All authors contributed to preparation of the manuscript.

## Additional information

**Competing interests:** The authors declare no competing financial interests.

