## [Peer Review File · Nature Communications]

Reviewers' comments:

Reviewer #1 (Remarks to the Author):

Review Report on "Mobile Microelectrode: Unified Label-Free Selective Cargo Transport" by A.M. Boymelgreen, T. Balli, T. Miloh, G. Yossifon.

This paper is demonstrating cargo transport by Janus particles using rich behaviors in electrokinetic effects such as ICEO (Induced Charge Electroosmosis) and DEP (Dielectric-phoresis). Authors found attraction of small particles (100-1000nm polystyrene) toward the gap between the metallic hemisphere of a Janus particle and the wall electrode under AC electric field. At low frequency, Janus particles move in the direction of polystyrene side, on the other hand at high frequency it moves in the direction of metal hemisphere. The former effect has been known as ICEO, while the latter effect has not been understood. The mechanism of the latter effect has been recently proposed by the same authors of this paper which is called self-dielectricphoresis (sDEP). The authors utilize the fact that the positive DEP and the negative DEP depend on external frequency and possibly on particle sizes for selectively attract or repel nano particles. Cargo particles can be selected by size and released by changing frequency of AC electric field. Paper is nicely written and well described by their own theory. I therefore recommend publication in Nature Communication. However, the following points should be clarified before publication.

In Figure 4 (a), The Clausius-Mossotti factor of target (300nm particles) and contaminant (1 μ m) are plotted as a function of frequency. The crossover frequencies at which the positive DEP (pDEP) changes into the negative DEP (nDEP) are different for target and contaminant. There seems to be the size dependence in the crossover frequency since the material are the same. However, in the reference 14 and 26 which are cited for the expression of Clausius-Mossotti factor in DEP theory, there is no size dependency in the crossover frequency. The author should explain why magenta curve and purple curve in Fig. 4(a) are different. At least in the supplementary information, they should write down the exact form of the expression of $\text{Re}(K)$ explaining how two curves are obtained with parameters used therein.

In Figure 4 (a), are plotted as a function of frequency. The same author published a theory for self-dielectricphoresis (sDEP) to account for the reversal of moving direction in Janus particle at high frequency (ref.5). It will be better to plot the theoretical curve of in Fig.4 (a) if it is available.

Although Janus particles can transport and release cargo selectively, their self-propelled motions are limited in a straight motion in general. How the Janus particles can transport cargo toward the target position? Is there any method to overcome this problem?

There are minor points.

Caption of Fig.4: The following sentences in (b) "(b) Operating condition ... (magenta and purple curves)." should move to (A).

Electrokinetics is a rather specific area for broad readers of Nature Communication. At least, pDEP and nDEP should be described shortly in the text.

In page 7, h should be defined.

Reviewer #2 (Remarks to the Author):

The idea to use the active transport of Janus particles as "mobile electrode", described in the manuscript sounds very exciting. The manuscript is well written. Generally speaking, originality of approach makes it possible to publish such manuscript in Nature Communications, but there are several aspects that definitely have to be addressed before final decision is done.

1. The impact of this approach/technique is not properly stated: while you mention the bottom up fabrication as an interesting application, please compare to the state of the art technique. (e.g. to which extent will it outperform the existing methods)?

2. The mechanism of electrophoretic/dielectrophoretic motion of colloidal particles in an applied electric field is known. Both plain silica and Janus particles will reveal the Motion, once field is applied, (both, AC and even DC). With respect to the motion and transportation: why "Janus" concept is necessary?

3. Please, could you specify better the "selectivity window" of the particles by size? (e.g. can particles selectively transport 900nm and 1µm beads). Clearly it is defined somewhere by frequency dependence of the dielectric constant, etc.

4. With respect to the Impact/Application again: Level of Nature communications would mean, on my opinion, to demonstrate the capability of the system to do something specific, apart from demonstrating the phenomena. At the Moment manuscript sounds to me like an exciting story without culmination.

After addressing aforementioned issues, once can consider the manuscript for publication.

Reviewer #3 (Remarks to the Author):

Manuscript review:

Title: " Mobile Microelectrodes: Unified Label-Free Selective Cargo Transport by Active Colloids"

Authors: Alicia M Boymelgreen, Tov Balli, Touvia Miloh, Gilad Yossifon

This paper proposes the use of metallodielectric Janus particles (with diameters around 10 microns) as carriers of smaller particles by using dielectrophoresis (DEP). The Janus particles are moved between parallel electrodes by application of an ac electric field. This applied electric field polarizes the Janus particle so that it attracts or repels smaller particles by DEP. Therefore, the Janus particles become mobile floating electrodes that can selectively pickup and release smaller particles by using DEP and changing the frequency of the applied signal. The voltage amplitude can control the rate of transport and load.

I think the idea of floating electrodes that can be moved at will is quite interesting. The authors have experimentally demonstrated the feasibility of this idea. The authors have characterized experimentally the properties of these carriers relative to the cargo. They have also developed a semi-quantitative theory that explains these properties. I think the manuscript deserves publication. Prior to this, the authors should take into account the following comments (second one is mandatory):

1. Although the carriers can be moved by the electric field, the actual direction of a carrier is, in principle, random. Could the authors comment on this and on ways of control this carrier motion direction?
2. In equation 0.1, page 4. The authors should say that $\nabla\chi$ is non-dimensional. C has units of voltage squared. More important, I think that there is a mistake in this expression and expression S6 from the supplementary material. From expression S2, the DEP force is the gradient of the following potential energy $U_{DEP} = -\pi b^3 \epsilon Re\{K\} E^2$ since $(\vec{E} \cdot \nabla)\vec{E} = \nabla E^2/2$, because the field is irrotational. According to page 2 of supplementary material, since $E \rightarrow -E_0 \nabla\chi/2$ for thin EDL, the DEP potential energy must be $U = -\pi b^3 \epsilon Re\{K\} E_0^2 \left(\frac{1}{2} \frac{\partial\chi}{\partial x}\right)^2$, i.e. it is proportional to the derivative of χ squared. The authors should correct the expression and, accordingly, the comparison with the theory together with plots of figures 2g and 3b.

Response to reviewers' comments:

Reviewer #1:

This paper is demonstrating cargo transport by Janus particles using rich behaviors in electrokinetic effects such as ICEO (Induced Charge Electroosmosis) and DEP (Dielectric-phoresis). Authors found attraction of small particles (100-1000nm polystyrene) toward the gap between the metallic hemisphere of a Janus particle and the wall electrode under AC electric field. At low frequency, Janus particles move in the direction of polystyrene side, on the other hand at high frequency it moves in the direction of metal hemisphere. The former effect has been known as ICEO, while the latter effect has not been understood. The mechanism of the latter effect has been recently proposed by the same authors of this paper which is called self-dielectricphoresis (sDEP). The authors utilize the fact that the positive DEP and the negative DEP depend on external frequency and possibly on particle sizes for selectively attract or repel nano particles. Cargo particles can be selected by size and released by changing frequency of AC electric field. Paper is nicely written and well described by their own theory. I therefore recommend publication in Nature Communication. However, the following points should be clarified before publication.

We thank the reviewer for supporting publication and clarified all comments.

In Figure 4 (a), The Clausius-Mossotti factor of target (300nm particles) and contaminant (1 μ m) are plotted as a function of frequency. The crossover frequencies at which the positive DEP (pDEP) changes into the negative DEP (nDEP) are different for target and contaminant. There seems to be the size dependence in the crossover frequency since the materials are the same. However, in the reference 14 and 26 which are cited for the expression of Clausius-Mossotti factor in DEP theory, there is no size dependency in the crossover frequency. The author should explain why magenta curve and purple curve in Fig. 4(a) are different. At least in the supplementary information, they should write down the exact form of the expression of $\text{Re}(K)$ explaining how two curves are obtained with parameters used therein.

We thank the reviewer for pointing this out. The size dependence arises from the dominance of surface conductance over the negligible material conductivity. We have elaborated on this in the introduction (page 3), the supporting material (page 2) and in the text discussing Figure 4 (page 10) and added additional references discussing this concept (Reference 18,19). We have clarified that the plots were obtained by experimentally obtaining the crossover frequency of the two different sized particles, and using this to determine the effective surface conductivity substituted into the expression for the Clausius-Mossotti factor (page 11).

In Figure 4 (a), are plotted as a function of frequency. The same author published a theory for self-dielectricphoresis (sDEP) to account for the reversal of moving direction in Janus particle at high frequency (ref.5). It will be better to plot the theoretical curve of in Fig.4 (a) if it is available. *In Reference 5, we focused on developing an expression for the critical frequency at which the Janus spheres reverse direction. The full frequency dispersion curve is complicated by competing effects in the low frequency domain and is still a work in progress. We have thus limited the plot to the experimental values here.*

Although Janus particles can transport and release cargo selectively, their self-propelled motions are limited in a straight motion in general. How the Janus particles can transport cargo toward the target position? Is there any method to overcome this problem?

We have reemphasized and expanded the suggestions for directed motion which include two methods

using AC electric fields; the control loop demonstrated by Mano et al in an identical system (Ref.14), hydrodynamic confinement (Ref.7). For applications where the frequency range prevents the usage of these methods, alternative methods such as passive boundary guidance (Ref.6) or the incorporation of a secondary mechanism such as magnetophoresis (Ref.15) could also be used(see Introduction page 3 and conclusion page 14).

There are minor points:

Caption of Fig.4: The following sentences in (b) “(b) Operating condition ... (magenta and purple curves).” should move to (A).

Done.

Electrokinetics is a rather specific area for broad readers of Nature Communication. At least, pDEP and nDEP should be described shortly in the text.

Defined in the introduction (page 3) and supporting materials..

In page 7, *h* should be defined.

Done.

Reviewer #2:

The idea to use the active transport of Janus particles as "mobile electrode", described in the manuscript sounds very exciting. The manuscript is well written. Generally speaking, originality of approach makes it possible to publish such manuscript in Nature Communications, but there are several aspects that definitely have to be addressed before final decision is done.

1. The impact of this approach/technique is not properly stated: while you mention the bottom up fabrication as an interesting application, please compare to the state of the art technique. (e.g. to which extent will it outperform the existing methods)?

In response to the reviewer's comments we have edited the introduction and conclusion to more clearly convey the impact of the approach/technique, what we have achieved and our view to the future. Essentially, the unified cargo carrier, which is the focus of this paper, is the first demonstration of the potential of this mobile microelectrode. When compared to other cargo transport mechanisms, our system has the advantage of:

- a. The propulsive and loading mechanisms are controlled by the same field –“unified”.*
- b. The loading is not restricted to “tagged” or surface functionalized cargo but could potentially be used for both organic and inorganic targets.*
- c. The choice of target can be varied in situ by altering the frequency of the applied field.*

At the same time, we recognized that beyond the specific application of cargo transport demonstrated herein, the JP has the ability to vary the spatio-temporal distribution of the electric field gradient from the “bottom up”- as opposed to standard approach of “top-down” photolithographic patterning. As a view to the future it is suggested this idea could be combined with directed motion and self-assembly to obtain flexible microfluidic systems with high resolution components that can be assembled/disassembled in situ (see conclusion).

2. The mechanism of electrophoretic/dielectrophoretic motion of colloidal particles in an applied electric

field is known. Both plain silica and Janus particles will reveal the Motion, once field is applied, (both, AC and even DC). With respect to the motion and transportation: why "Janus" concept is necessary?

A uniform sphere in a uniform AC field is not expected to reveal motion since due to the zero time average, a net effect requires symmetry breaking. We have clarified however, that while the Janus effect is critical for active colloidal mobility, it is less critical for the target loading as a homogeneous polarizable sphere could also function as a freely suspended (but immobile) floating electrode, however, with weaker electric field gradients as in the former JP wherein the discontinuity of the metallo-dielectric interface intensifies the induced gradients (page 3).

3. Please, could you specify better the "selectivity window" of the particles by size? (e.g. can particles selectively transport 900nm and 1μm beads). Clearly it is defined somewhere by frequency dependence of the dielectric constant, etc.

We have clarified the relationship between the selectivity window and the Clausius-Mossotti factor in the text adjacent to Figure 4 (page 11). The size dependence arises from the dominance of surface conductance over the negligible material conductivity. We have elaborated on this in the introduction (page 3), the supporting material (page 2) and in the text discussing Figure 4 (page 10) and added an additional references discussing this concept (Reference 18,19).

4. With respect to the Impact/Application again: Level of Nature communications would mean, on my opinion, to demonstrate the capability of the system to do something specific, apart from demonstrating the phenomena. At the Moment manuscript sounds to me like an exciting story without culmination. *The paper clearly demonstrates the application of the "mobile microelectrode" to dynamically select cargo and transport it without the need for pre-labelling, all while using a single external control. Such a unified label-free selective cargo transport by active colloids is demonstrated herein for the first time. In addition, we note that such a system constitutes a paradigm shift relative to the immense body of work on dielectrophoresis wherein the electrodes are fixed.*

After addressing aforementioned issues, once can consider the manuscript for publication.

We have addressed all issues raised by the referee and appreciate his positive approach towards publication.

Reviewer #3:

This paper proposes the use of metallodielectric Janus particles (with diameters around 10 microns) as carriers of smaller particles by using dielectrophoresis (DEP). The Janus particles are moved between parallel electrodes by application of an ac electric field. This applied electric field polarizes the Janus particle so that it attracts or repels smaller particles by DEP. Therefore, the Janus particles become mobile floating electrodes that can selectively pickup and release smaller particles by using DEP and changing the frequency of the applied signal. The voltage amplitude can control the rate of transport and load.

I think the idea of floating electrodes that can be moved at will is quite interesting. The authors have experimentally demonstrated the feasibility of this idea. The authors have characterized experimentally the properties of these carriers relative to the cargo. They have also developed a semi-quantitative theory that explains these properties. I think the manuscript deserves publication. Prior to this, the authors should take into account the following comments (second one is mandatory):

1. Although the carriers can be moved by the electric field, the actual direction of a carrier is, in principle, random. Could the authors comment on this and on ways of control this carrier motion direction?

We have reemphasized and expanded the suggestions for directed motion which include two methods using AC electric fields; the control loop demonstrated by Mano et al in an identical system (Ref.14), hydrodynamic confinement (Ref.7). For applications where the frequency range prevents the usage of these methods, alternative methods such as passive boundary guidance (Ref.6) or the incorporation of a secondary mechanism such as magnetophoresis (Ref.15) could also be used(see Introduction page 3 and conclusion page 14).

2. In equation 0.1, page 4. The authors should say that $\nabla\chi$ is non-dimensional. C has units of voltage squared. More important, I think that there is a mistake in this expression and expression S6 from the supplementary material. From expression S2, the DEP force is the gradient of the following potential energy $U_{DEP} = -\pi b^3 \epsilon Re\{K\} E^2$ since $(\vec{E} \cdot \nabla)\vec{E} = \nabla E^2/2$, because the field is irrotational. According to page 2 of supplementary material, since $E \rightarrow -E_0 \nabla\chi/2$ for thin EDL, the DEP potential energy must be $U = -\pi b^3 \epsilon Re\{K\} E_0^2 \left(\frac{1}{2} \frac{\partial\chi}{\partial x}\right)^2$, i.e. it is proportional to the derivative of χ squared. The authors should correct the expression and, accordingly, the comparison with the theory together with plots of figures 2g and 3b.

We have added the dimensions of C .

We thank the referee for pointing out a misprint in eq.(S5) resulting in the new definition of $\tilde{\chi}$. We have elaborated in the supporting materials that the expression obtained by the reviewer is equivalent to the leading order expression of eq.(S7).

REVIEWERS' COMMENTS:

Reviewer #1 (Remarks to the Author):

I have read the resubmitted manuscript and response letter.

My questions to the authors have been properly replied. They added elaborated explanations in the main text, supplementary materials, and in the references. Based on these correspondence and revision, I think the paper deserve publication.

Reviewer #2 (Remarks to the Author):

After checking the manuscript and Response to Reviewers file, i have rather positive Impression about the authors dedication. Some points (e.g. 1 and 4, Respose letter, Reviewer 2) are, however, still stay a bit unclear: answers are not really precise.

Point 1: bottom up fabrication and state of the art technique.

I agree that the mechanism of cargo transport is demonstrated for the first time and is new for the janus particle-cargo transport community.

But since you try to address pretty important problem (not only scientific, but also engineering one), and because you want to publish the story in Nature Communication, i ask you to extend your vision beyond the field of only Janus particles and micromotors. Do not forget about soft robotics, and even fast growing field of additive manufacturing. In which sense (parameters: scale, speed, time, precision????) your technique can represent a breakthrough, for more broader field?

Point 4. Impact/Applications. Authors answer "The paper clearly demonstrates the application of the "mobile microelectrode" ...". I would disagree. Paper demonstrates the "possibiity to apply", but not an application. These are two different things. Since transport of cargo can be fully addressable (and you claim it), try to show that you can build at least a straight line of transported particles using your technique. This will be a clear application.

Overall, i have a positive impression about the mansucript and would be happy to see it published in Nature communications, after addressing the remaining issues.

Reviewer #3 (Remarks to the Author):

After reading the authors' reply to my previous comments, I recommend publication.

Response to reviewers' comments:

Reviewer #1:

“I have read the resubmitted manuscript and response letter. My questions to the authors have been properly replied. They added elaborated explanations in the main text, supplementary materials, and in the references. Based on these correspondence and revision, I think the paper deserve publication.”

We thank the referee for supporting publication.

Reviewer #2:

“After checking the manuscript and Response to Reviewers file, I have rather positive Impression about the authors dedication. Some points (e.g. 1 and 4, Response letter, Reviewer 2) are, however, still stay a bit unclear: answers are not really precise.”

Point 1: bottom up fabrication and state of the art technique.

I agree that the mechanism of cargo transport is demonstrated for the first time and is new for the Janus particle-cargo transport community. But since you try to address pretty important problem (not only scientific, but also engineering one), and because you want to publish the story in Nature Communication, I ask you to extend your vision beyond the field of only Janus particles and micromotors. Do not forget about soft robotics, and even fast growing field of additive manufacturing. In which sense (parameters: scale, speed, time, precision???) your technique can represent a breakthrough, for more broader field? “

We thank the referee for pointing this out and we have accordingly expanded on the potential of this mechanism for soft robotics and additive manufacturing. Specifically, we have expanded the discussion on the broader impact of the technology in the last paragraph of the discussion to suggest how the current carrier could be used for bottom up (or additive) manufacturing and enumerated the advantages vis a vis current state of the art top-down photolithography, specifically – reduced fabrication cost of the microfluidic chip, increased feature resolution which is not restricted to 2D planar geometries and the potential to dynamically assemble/disassemble these structures in situ. We have added a reference on the use of additive manufacturing in microfluidics (Ref.39). We have also suggested that as well as the previously suggested building blocks for smart materials, the frequency dependent selectivity could be used to build complex soft actuators for soft robotic applications as well as a review paper discussing the use of Janus particles in similar applications (Ref. 41).

“Point 4. Impact/Applications. Authors answer "The paper clearly demonstrates the application of the “mobile microelectrode”...". I would disagree. Paper demonstrates the "possibiity to apply", but not an application. These are two different things. Since transport of cargo can be fully addressable (and you claim it), try to show that you can build at least a straight line of transported particles using your technique. This will be a clear application.”

Following this comment made by the referee we added an explicit suggestion for the possibility of using the cargo carrier for the specific application of sample analysis wherein, when combined with directed motion the carrier could be used to transport a selected target to a secondary location for further analysis. We have included an additional reference discussing the use of Janus micromotors in such systems (Ref.36). As we have made no claims regarding directed motion, the further experiments suggested by the reviewer are beyond the scope of the current paper.

“Overall, I have a positive impression about the mansucript and would be happy to see it published in Nature communications, after addressing the remaining issues.”

We thank the referee for this positive recommendation and have addressed all remaining issues.

Reviewer #3:

After reading the authors' reply to my previous comments, I recommend publication.

We thank the referee for supporting publication.